# Integrating Programmatic Expertise from across the US and Canada to Model and Guide Leadership Training for Graduate Students in Sustainability

Nicole Motzer [1,†], Aleta Rudeen Weller [2,*,†], K Curran [3], Simon Donner [4], Ronald J. Heustis [5], Cathy Jordan [6], Margaret Krebs [7], Lydia Olandar [8], Kirsten Rowell [9], Linda Silka [10], Diana H. Wall [2,11] and Abigail York [12]

1 National Socio-Environmental Synthesis Center, Annapolis, MD 21401, USA; nmotzer@sesync.org
2 School of Global Environmental Sustainability, Colorado State University, Fort Collins, CO 80523, USA; diana.wall@colostate.edu
3 The Pew Charitable Trusts, Washington, DC 20004, USA; kcurran@pewtrusts.org
4 Department of Geography, Institute for the Oceans and Fisheries, University of British Columbia, Vancouver, BC V6T 1Z2, Canada; simon.donner@ubc.ca
5 Department of Biological Chemistry and Molecular Pharmacology, Blavatnik Institute, and Office of Graduate Education, Harvard Medical School, Boston, MA 02115, USA; ronald_heustis@hms.harvard.edu
6 Institute on the Environment, University of Minnesota, St. Paul, MN 55108, USA; jorda003@umn.edu
7 Earth Leadership Program, Woods Institute for the Environment, Stanford University, Stanford, CA 94305, USA; mkrebs@stanford.edu
8 Nicholas Institute for Environmental Policy Solutions, Duke University, Durham, NC 27708, USA; lydia.olander@duke.edu
9 Environmental Studies and Research and Innovation Office, University of Colorado Boulder, Boulder, CO 80309, USA; kirsten.rowell@colorado.edu
10 Senator George J. Mitchell Center for Sustainability Solutions and School of Economics, University of Maine, Orono, ME 04469, USA; silka@maine.edu
11 Department of Biology, Colorado State University, Fort Collins, CO 80523, USA
12 School of Human Evolution and Social Change, Arizona State University, Tempe, AZ 85287, USA; abigail.york@asu.edu
* Correspondence: aleta.weller@colostate.edu; Tel.: +1-970-492-4160
† Co-first author.

**Abstract:** It is critical that future sustainability leaders possess the skills and aptitudes needed to tackle increasingly 'wicked' challenges. While much has been done to identify this need, inadequate Leadership Training for graduate students in Sustainability (LTS) continues to plague even the most highly-resourced institutions. Collectively, the authors of this paper represent the small yet growing number of LTS programs across the United States and Canada working to close this training gap. In this paper, we describe the integrative approach we took to synthesize our collective knowledge of LTS with our diverse programmatic experiences and, ultimately, translate that work into concrete guidance for LTS implementation and design. We present a framework for the suite of key LTS aptitudes and skills yielded by our collaborative approach, and ground these recommendations in clear, real-world examples. We apply our framework to the creation of an open-access curricular database rich with training details, and link this database to an interactive network map focused on sharing programmatic designs. Together, our process and products transform many disparate components into a more comprehensive and accessible understanding of what we as LTS professionals do, with a view to helping others who are looking to do the same for the next generation of sustainability leaders.

**Keywords:** sustainability; leadership; graduate program; higher education; aptitudes; skills; interdisciplinary; training

## 1. Introduction

The immensity and complexity of modern global challenges have fundamentally altered sustainability researchers' and professionals' roles and responsibilities [1] and

demanded of graduate students—our future sustainability leaders—an unprecedented pairing of leadership and scientific mastery. Yet, with conventional modes of technical and disciplinary graduate training dominating the higher education landscape, graduate curricula rarely prioritize non-technical and interpersonal leadership skills, like critical reflection and communication, that are essential to leading the fight against pressing sustainability issues [2–4]. It is instead common for declarative, or content, knowledge within the domains of individual disciplines to take precedence [5,6].

When students are trained through siloed or lecture-based approaches alone, they are not only denied exposure to the diversity of experiences and expertise inherent to sustainability issues [7] but may also be prevented from developing the skills and aptitudes that cross-cutting, dynamic, real-world challenges require the most [8–10]. In such cases, it is not surprising that graduate students are often dissatisfied with how their graduate training has prepared them to engage in problem solving and collaboration [11], or that they can struggle to put into practice the core dimensions of sustainability science [12], namely: interdisciplinary research, stakeholder engagement, and translating knowledge into solutions [13,14].

Becoming a sustainability leader requires both content knowledge and practical skills [15]. Following Visser and Courtice [16], we define a sustainability leader as "someone who inspires and supports action towards a better world" (p. 2). Put another way, sustainability leaders must gain expertise in both scientific thinking and taking real-world actions [17]. As Shriberg and Harris attest, "we cannot simply tell students to go . . . be a sustainability leader without providing the structure and skills training for success" [18] (p. 154).

For programs and institutions that do strive to provide a more comprehensive and skills-based approach for sustainability leaders-in-training, another set of challenges can interfere: (1) over-reliance on already burdened faculty or on instructors without the appropriate expertise [19]; (2) limited guidance for operationalizing the wide array of often vague and disjointed sustainability competencies [20–22]; and (3) overall lack of documentation, communication, and exchange between existing efforts around experiences and best practices [9,23], which results in little organized guidance as to program design and implementation [24].

This paper captures the process and results of an international, interdisciplinary, and multi-institutional network initiative that sought to address all three of these hurdles in the context of Leadership Training for graduate students in Sustainability (LTS). Stemming from a working meeting at the US National Socio-Environmental Synthesis Center (SESYNC) in February of 2020, the curricular and network mapping approach we ultimately present pairs a set of rigorously co-developed LTS competencies (i.e., aptitudes) with in-depth documentation of tested training activities and programmatic design details for seven established LTS programs across North America. In addition to serving as both a foundation from which other LTS programs and professionals can learn, self-assess, and build, the interlinked products described here demonstrate the benefits of enhanced inter-program collaboration and bolster abilities to "teach what we preach" [25] (p. 1725) to tomorrow's sustainability leaders.

## 2. Literature Review

The movement towards filling gaps in sustainability education has accelerated in recent years with growing recognition that our students are key to achieving a more sustainable future [10]. There is no longer any lack of classroom resources to teach about sustainability, with prominent examples like UNESCO's Teaching and Learning for a Sustainable Future, the Association for the Advancement of Sustainability in Higher Education's Disciplinary Associations Network for Sustainability, and the National Council for Science and the Environment all providing a wealth of material. However, growing recognition around what it takes to effectively tackle sustainability challenges, including managing urgency, rapid change, and uncertainty [26], has shifted priorities away

from teaching sustainability as a classroom concept to instead honing the particular skills students need in order to be transformed into sustainability leaders [9,18].

This transition has prompted the creation of countless courses, workshops, and collaborations under the umbrella of sustainability training. Still, debate remains regarding what form such training should take [19,27]. The US National Academies project "Strengthening Sustainability Programs and Curricula at the Undergraduate and Graduate Levels" is just one recent example of an effort to make strides in this space (This effort is chaired by ANGLES member Dr. Anne Kapuscinski of the University of California Santa Cruz). A parallel transition can be seen in scholarship over the last decade, most notably in the myriad lists of competencies put forth intending to crack the code of sustainability training success (e.g., [5,28–31]). Wiek and colleagues, for example, propose the following competency groups—and the ability to integrate them—as integral to solving sustainability issues: systems thinking, anticipatory, normative, strategic, and interpersonal [22]. In another example, Rieckmann argued for recognition of the importance of the following competencies: systems thinking and handling of complexity, anticipatory thinking, critical thinking, acting fairly and ecologically, cooperation in groups, participation, empathy and change of perspective, interdisciplinary work, communication and use of media, planning and realizing innovative projects, evaluation, and ambiguity and frustration tolerance [32].

Proposed competencies have been instrumental in guiding educational expectations and outcomes [33]. However, listing competencies is insufficient when it comes to bridging the gaps between conventional graduate education and sustainability scientists' and professionals' updated roles and responsibilities. Conceptually linking related competencies [22] and matching particular competencies with the teaching and learning methodologies—like matching group collaboration with project and problem-based learning (PPBL) [5,30]—that are most likely to convey those competencies are some of the ways in which scholars have moved past competency identification and toward acquisition. Evans takes such pairings one step further, offering descriptive examples of the knowledge, skills, and attitudes that may signal whether a competency has been learned [28]. For example, instructors attempting to convey systems thinking competence may monitor whether students exhibit sensitivity to context. Even with such progress, though, conceptual recommendations for sustainability skills training can fall short if uncoupled from practical demonstrations and real-world learning experiences [21,34].

Recognizing this, Shriberg and Harris describe how a systems thinking framework and PPBL pedagogy together inform the activities and design of a campus-based course aimed at cultivating sustainability leadership and organizational change [18]. Their evaluation results shed light on the key lessons learned, including the value of spending less time on concepts and more time learning skills in action. Wiek et al. similarly model how to operationalize sustainability competencies using PPBL in a real-world university program [9]. In particular, they provide detailed descriptions of how courses, workshops, and projects incorporate the PPBL learning pedagogy, including how their model adapts to different learning styles, learning settings, and student-world interactions. They pull back the curtain further in order to discuss specific types of support, like a community-university liaison on staff, that make their training model possible.

In other cases, Gardiner and Rieckmann draw from their university course to model how reflective journaling can help students to navigate multiple, uncertain futures and to build anticipatory competence specifically [35], while Newman-Storen describes how a master's program encourages the development of change agents and creative sustainability solutions via student environmental art projects that facilitate creative thinking [36]. More recently, Roy et al. documented their multi-institutional course as an approach for conveying competencies like interdisciplinarity, stakeholder engagement, and problem-solving [6]. They explain how to operationalize core tenets through student-instructor co-creation, student leadership, and class activities ranging from case study research to think-pair-share exercises.

Despite the existence of several examples in the literature which offer important insights, concrete guidance that better situates sustainability competencies within curricular development and program design has remained the exception rather than the rule, especially for LTS [20]. As the label suggests, LTS ushers sustainability competencies into the realm of leadership for sustainability. Leadership for sustainability is distinct from other more traditional models of leadership in several defining ways. First, it occurs within the specific context of highly complex and globalized sustainability challenges, and with the specific goal of a socially, economically, and environmentally sustainable future for the benefit of all [16]. 'Wicked' sustainability challenges and goals require leaders to have a particular suite of capacities in order to effectively respond and adapt to unpredictable, unprecedented, and, at times, unsolvable issues [37]. Second, sustainability leadership is multi-directional and non-hierarchical [16]. Lines between leaders and non-leaders are blurred, and anyone is thought to be capable of becoming a sustainability leader, since sustainability challenges are increasingly widespread and indiscriminate [37,38]. Third, it is common for sustainability leaders to apply a variety of leadership styles based on the matter at hand, particularly those styles that are in line with the 'new leadership era' [39] (p. 4). For example, inclusive leadership may be most suitable if greater community input is desired when designing adaptation plans [40], whereas complexity leadership may be more appropriate when addressing a systems-level issue that crosses numerous sectors and borders [41].

Higher education has enormous potential to prepare graduate students for effective sustainability leadership. As members and affiliates of ANGLES (A Network for Graduate Leadership in Sustainability), the authors of this paper collectively represent the small yet growing number of institutions and programs across the United States and Canada that are working to tap into this potential by enhancing and expanding LTS. We argue that a community- and practice-driven approach that integrates the collective and institutional knowledge of diverse experts and programs with common goals but distinct approaches, and that illuminates what leadership for sustainability development looks like in varied university settings, is critical to advancing LTS. We seek to provide such an approach with this paper. In the sections that follow, we describe the synthesis process and activities through which our initial program and knowledge integration took place. Next, we present the resulting co-produced list of LTS competencies (or "aptitudes") as a framework for program implementation and design, and embed those aptitudes and skills in real-world examples of how each might be taught and learned. We then outline how we linked training details to in-depth information around programmatic design via the creation of an open-access curricular database and an interactive network map. Finally, we reflect on the outcomes of our process and conclude by discussing the main findings and future directions for those working in the LTS space.

## 3. Process

The ANGLES network was founded in 2017 in order to mitigate isolation among LTS programs operating across the US and Canada and, instead, harness the collective energy and expertise of a diverse consortium. Three years after the initial meeting, convened by Stanford's Leopold Leadership Program and Institute on the Environment at University of Minnesota, ANGLES members and affiliates met for three days at the US National Socio-Environmental Synthesis Center (SESYNC) in Annapolis, Maryland, USA. The meeting, which we describe in-depth below, followed a synthesis approach in order to facilitate the integration and exchange of individual programs and expertise into comprehensive guidance and actionable models for LTS [42].

A total of 13 individuals attended the meeting, including each co-author along with an expert facilitator. Individuals were invited to participate for different, albeit complimentary, reasons: some direct long-running LTS programs; some are noted leadership experts; some have prior experience fostering communities of practice; and some train graduate students in fields other than sustainability and thus represent an outside perspective. As a measure

of collective expertise, attendees brought to the meeting a combined total of 80 years in LTS, and prior experience working with nearly 2500 graduate students or other scholars via LTS initiatives over time. Attendees specialized in skills ranging from interdisciplinary team science, to science communication, to stakeholder engagement and balancing science and advocacy. For those who directed LTS programs, none approach LTS in quite the same way; some operated as campus-based cohorts, some virtually connected with students across universities, and others focused on standalone seminars and retreats. The lenses through which attendees provided LTS spanned ocean sciences, JEDI (justice, diversity, equity and inclusion), public policy, and more. All attendees were previously or actively involved in the ANGLES network.

Process-wise, the meeting was built around an iterative series of full group discussions, focused breakout group discussions, and individual activities (as illustrated in Figure 1). Expert facilitation and expert elicitation were the predominant methods of choice [43]. Day 1 entailed surveying the existing sustainability leadership and LTS landscapes in order to locate gaps between trainings ideals and realities and defining an initial list of core LTS aptitudes and skills. An example of a guiding question for the day included: "What makes LTS different? Why isn't it as simple as replicating existing training models?" We began developing the list of LTS aptitudes by working first in small groups, and then, as the full group, refining what we considered to be the individual leadership skills that sustainability challenges demand. In other words, we identified what we as experts and practitioners in the LTS space think is required of those who must manage increasingly complex and ever-evolving sustainability problems. We then grouped those skills into overarching categories that we labeled "aptitudes".

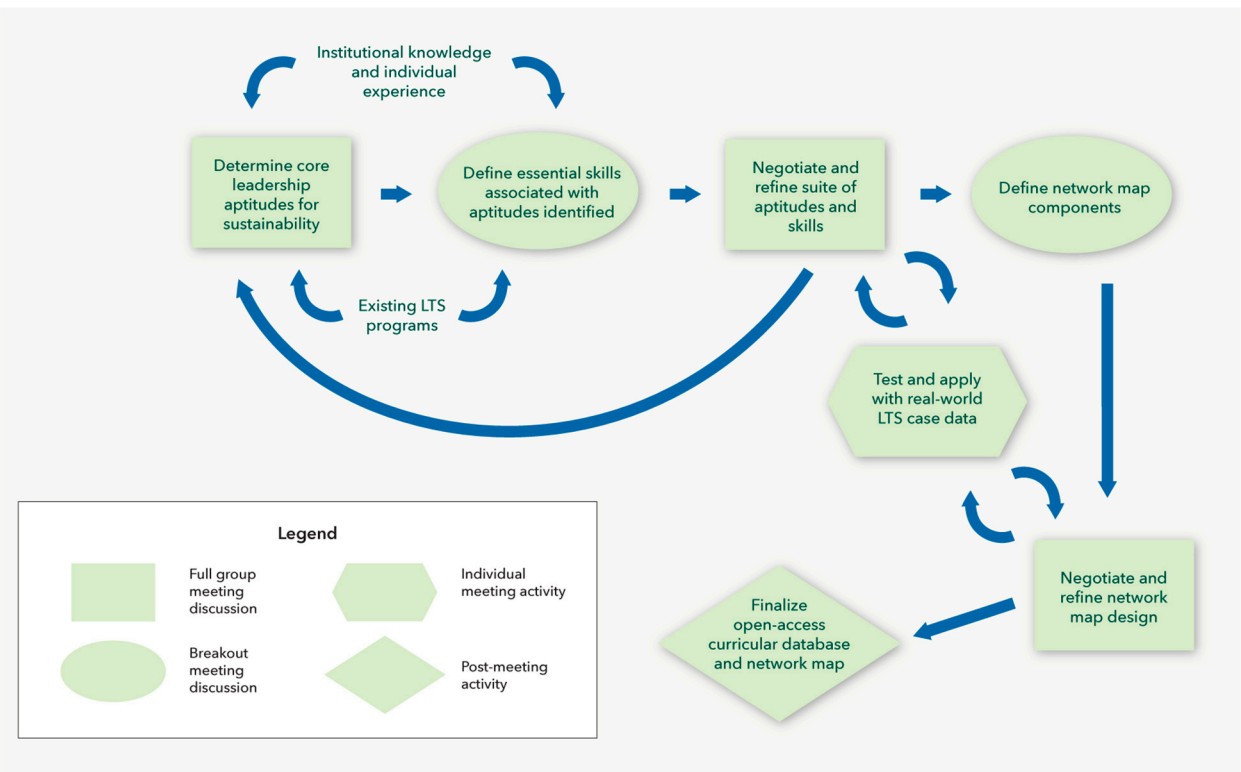

**Figure 1.** Iterative and integrative working group process.

A motivating question for Day 2 was: "Based on the assets and expertise we each have to offer, what might we build together? What could we collectively do that we couldn't if acting alone?" As such, Day 2 consisted of finalizing the initial aptitudes list into a framework, but also brainstorming and testing strategies for integrating our diverse programmatic philosophies, resources, and activities into a comprehensive model and shareable platform. We solicited from meeting attendees detailed examples of how they and their programs interpret and implement LTS aptitudes and skills. We captured this information via a webform, which enabled individuals to upload each of their programs' core activities and tag them with all the corresponding skills. In addition to aptitudes and skills, webform entries asked for descriptions of how exactly the activity works to convey the corresponding skills, activity or event frequency and duration, and for any activity materials or resources contributors would like to publicly share.

We also began to document the forward-facing aspects of each of our programs before expanding into a discussion of their innermost workings. This operated as a first step in linking training data with additional programmatic details. We drew from our own experiences and insights surrounding what makes LTS programs possible from administrative and institutional standpoints. We also discussed which programmatic aspects would be most useful for others to know when launching or designing programs of their own. We used our own program data to populate, discuss, and revise the structural components and logical basis of the database in order to accommodate the diverse sample of network programs (and, eventually, programs operating outside of the network). As we noticed overlaps or gaps in the way one another's programs responded to the core map elements, we refined the categories. Data entries followed a webform approach similar to that which was used for the aptitudes database described above. Day 3 continued with finalizing unfinished activities and discussions from Day 1 and 2 and laid the foundations for post-meeting activities and outcomes.

## 4. Outcomes

### 4.1. Aptitudes List and Curricular Database

The aptitudes and skills that emerged from discussions in our meetings were based on our combined programmatic experiences, as well as derived from our knowledge of previously published sustainability competencies. Outcomes were also shaped by a consideration of the need to be relevant to all graduate student trainees, regardless of whether their sustainability careers take an academic, practitioner, or alternative route. The final framework included a total of seven aptitudes and 48 corresponding skills (see Appendix A). The seven aptitudes are: (1) fostering belonging, equity, diversity and justice; (2) building emotional intelligence; (3) collaborating for impact; (4) communicating and engaging; (5) strategic thinking and planning; (6) working productively and effectively; and (7) making work matter (see Figure A1 in Appendix B). The framework and training activities data we collected, both on-site and following the meeting, laid the foundation for a searchable, open-access curricular database that is now housed on the ANGLES website ("Skills and Aptitudes Database" https://anglesnetwork.com/angles-aptitudes/).

Below, we provide descriptions for each aptitude that include why and how it belongs in LTS. After listing the corresponding skills, we provide an example of how the aptitude and skills have been interpreted and applied. Visitors to our online curricular database can access further information about each of these programs, along with dozens more examples of how network members and affiliates put LTS aptitudes and skills into practice.

1. Fostering Justice, Equity, Diversity, and Inclusion (JEDI)

Sustainability leaders must recognize that durable sustainability solutions come from the inclusion of diverse backgrounds, needs, expertise, and ways of knowing, and must prioritize equity in both process and outcomes. Sustainability leaders themselves must also represent the diversity of the human experience in the regions and communities where they seek to make change. In the context of sustainability, we define justice as dismantling barriers to resources and opportunities in society so that all individuals and

communities can live a full and dignified life. Equity means allocating resources to ensure everyone has access to the same resources and opportunities. Diversity relates to the different ways in which we experience systemic advantages or encounter systemic barriers to opportunities. Inclusion happens when a sense of belonging is fostered by centering, valuing, and amplifying the voices, perspectives and styles of those who experience more barriers based on their identities. Designing LTS around the critical reflection of systemic barriers and biases, inclusive decision-making, and equitable problem-solving is essential to building a just and sustainable future for all.

Associated Skills:

1.1 Understanding the landscape of JEDI
1.2 Appreciating the why of JEDI
1.3 Approaching your work and making decisions informed by JEDI
1.4 Fostering belonging and empowering others

Example: Global Sustainability Scholars Program at the University of Colorado Boulder

With the goal of building a lifelong network of diverse sustainability professionals trained in the inclusive practice of transdisciplinary research, JEDI is built into the backbone of the Global Sustainability Scholars Program. It is woven throughout every activity rather than existing as a specific learning objective of any one training or workshop. The program trains and connects graduate students from underrepresented groups (As defined by the National Science Foundation: Women, persons with disabilities, and underrepresented minority groups—blacks or African Americans, Hispanics or Latinos, and American Indians or Alaska Natives.) from universities across the United States to international sustainability projects. Cohort-based virtual programming focuses on building an identity within the field and changing the culture of sustainability research towards valuing learning that takes place as a result of diverse perspectives and collaborations. As part of the application process, applicants write an essay on the role of diversity and equity in the field of sustainability in order to demonstrate their understanding of the need for JEDI-based thinking and interventions. A mix of workshops and trainings that span qualitative methods, science communication, and professional development skills are crosscut with conversations about power dynamics in research, and how to conduct research in line with JEDI principles. Fellows are provided with opportunities to enact such principles in real-world contexts by engaging in cutting-edge transdisciplinary research. The program's formal and informal interactions with diverse mentors and peers promote opportunities for students to build community and support one another on their way to becoming global sustainability leaders.

2. Building Emotional Intelligence

Sustainability leaders must be able to acknowledge, value, and empathize with others in order to effectively represent them and successfully champion key issues. Self-awareness and the appropriate expression of one's own emotions, beliefs, and skills is essential when it comes to inspiring others and building resilience in the face of setbacks. LTS models that prioritizes emotional intelligence skills will help trainees to identify their own areas that are in need of personal and interpersonal growth throughout their leadership journeys. Acquiring such skills will also prepare them to more effectively manage both the internal and external tensions that arise when dealing with 'wicked' problems.

Associated Skills:

2.1 Building and maintaining personal integrity
2.2 Recognizing your own strengths and weaknesses
2.3 Recognizing your own values and motivations
2.4 Having an empathetic mindset
2.5 Valuing other's strengths, weaknesses, and values
2.6 Aligning your personal values and intent with the actions and strategy you choose
2.7 Rebounding from failure
2.8 Identifying your leadership style

Example: The Boreas Leadership Program at the Institute on the Environment, University of Minnesota

For a number of years, the Boreas Leadership Program offered an annual workshop series for graduate students which focused on personal and interpersonal mastery; the idea that self- and other-awareness are the foundations of effective leadership. With the aim of helping graduate students cultivate a greater understanding of human emotions and the importance of emotional fluency in leadership success, the series began with participants completing the Emotional Quotient Inventory (EQi), a self-report measure used to assess emotional intelligence and social competencies ("Emotional Quotient Inventory" http://www.eiconsortium.org/measures/eqi.html, accessed on 13 March 2021). The Boreas Program coordinator met with participants individually to facilitate understanding and the application of their emotional intelligence profile, including their strengths and weaknesses, potential leadership derailers, strategies for turning emotional intelligence into emotional effectiveness, and professional development planning (*skills 2.2–2.4, 2.6, 2.8*). The EQi served as a touchstone for participants throughout the remainder of the series, with additional workshops and coaching sessions provided for graduate students ranging in focus from developing intercultural competence (*skill 2.5*), mindfulness and developing and telling the story of what you do and why you do it (*skills 2.3, 2.6*) to negotiating controversy, difference, conflict, and difficult conversations (*skill 2.5*). For example, The Mindful Leader workshop paired evidence-based information about the neuroscience of empathy with practical mindfulness exercises focused on identifying sources of stress and their manifestation in the body in order to convert experiences of stress into fuel for achieving goals.

3.  Collaborating for Impact

Working in teams is essential for sustainability leaders, particularly when the urgency and complexity of sustainability challenges transcend any one scientific discipline, political party, sector, community, region or nation. Furthermore, achieving sustainability goals demands of sustainability leaders the novel integration of disparate resources, understandings, and experiences. LTS must provide applied opportunities to practice working in teams, in which valuing diverse contributions and finding common ground are rewarded.

Associated Skills:

3.1  Understanding and applying conflict resolution and negotiation skills
3.2  Understanding team interactions and establishing shared visions, norms, processes, and trust
3.3  Synthesizing and drawing connections among disparate ideas, information, theories, methods, evidence, and bodies of work
3.4  Appreciating diverse views, priorities, values, and epistemologies
3.5  Leveraging differences for improved lines of inquiry and problem solving
3.6  Empowering others by recognizing their skills and expertise
3.7  Co-developing and delivering outcomes

Example: The Earth System Science for the Anthropocene at Arizona State University

The Earth System Science for the Anthropocene is a cohort- and campus-based transdisciplinary graduate student network motivated by the understanding that no single discipline or knowledge system can adequately address modern human-environment challenges. Students are trained through a "basket-weaving" model, where a variety of experiences beyond the disciplinary foundations of each individual students' masters or doctoral program are woven together. The network's Immersive Team Science Experience is one such experience that provides students with a platform to creatively explore and combine interdisciplinary knowledge with community partners; practice ethical knowledge co-production; and develop feasible solutions through novel problem framings. A half-day preparatory workshop prepares students for the realities of interdisciplinary research and co-production (*skills 3.1, 3.2, 3.3, 3.4*). In the field, students engage with

identified communities and draw from instructor-assembled research products, such as associated qualitative and quantitative datasets, published research, and social or news media, to begin to collaboratively tackle specific sustainability problems (*skills 3.2, 3.3, 3.4*). Afterwards, teams participate in facilitated discussions to reflect on the lessons learned in conflict resolution, communication and the ethical challenges inherent in co-produced team science (*skills 3.1, 3.2, 3.5, 3.7*). Students who have completed one Immersive Team Science Experience are asked to share their story as a panelist in the following cohort's preparatory workshop to help incoming students learn and prepare (*skills 3.1, 3.2, 3.5, 3.6*).

4.   Communicating and Engaging

It is imperative that sustainability leaders not only understand how to effectively tell their audience or peers something they need to know, but to also sincerely listen to and value the wide array of perspectives, positions, and priorities that are embedded in complex sustainability issues (especially regarding opposing points of view). It is insufficient to simply better hone one's own message. Layering LTS with nested skills aimed at giving voice to and fostering exchange between diverse individuals and groups is prioritized over teaching individual-focused, unidirectional communication skills.

Associated Skills:

4.1   Asking good questions and having good conversations
4.2   Listening well and active listening
4.3   Giving and receiving feedback
4.4   Offering opposing points of view respectfully
4.5   Knowing your audience and tailoring your communication accordingly
4.6   Effectively presenting and conveying information

Example: The Sustainability Leadership Fellows program at Colorado State University's School of Global Environmental Sustainability

The Sustainability Leadership Fellows program is a year-long, cohort-based program for PhD students and postdoctoral fellows working in sustainability science at CSU. Science communication and engagement are some of the core tenets of the fellowship, which utilizes different types of communications training, layered throughout the year, in order to help future sustainability leaders not only compellingly convey information, but also engage more effectively in multi-directional communication and understanding. Fellows first participate in a two-day science communication workshop run by COMPASS science communication specialists, where fellows intensively interact and practice with expert journalists. This helps fellows to hone their message and identify effective science narratives through storytelling (*skills 4.3, 4.5, 4.6*). To transition beyond outward-focused communication, this initial workshop is later complemented with a half-day workshop on communicating with potentially skeptical audiences. The short course uses lectures, small group scenarios, and discussions to help fellows better understand (and truly listen to) their audience, discern beliefs and uncover shared values, and identify what they might learn from opposing points of view (*skills 4.1, 4.2, 4.4, 4.6*). Throughout the year, fellows also write a peer-reviewed blog post, published on the School's sustainability blog, and peer review the post of another to practice compelling narrative development and giving and receiving feedback outside their discipline (*skills 4.3, 4.6*). These activities help fellows to gain a broader world-view for how they might communicate, understand, and learn from differing points of view, stakeholder needs, and interdisciplinary partners.

5.   Strategic Thinking and Planning

Sustainability leaders face challenges that have no single solution, but often involve many possible responses with varying tradeoffs, insufficient information, and urgent timelines. Accounting for high uncertainty and complex interdependencies requires a clear vision, creative approach, and a means of measuring progress against goals. LTS must make room for skills that enable assessment and the implementation of new ideas and strategic appraisals of actions taken towards continually evolving situations.

Associated Skills:

5.1 Comfort with and straddling the frontiers of ambiguity
5.2 Brainstorming, visioning, and scenario planning
5.3 Risk and decision analysis, and decision-making under uncertainty
5.4 Seeing the big picture and thinking at the systems level
5.5 Evaluating, adapting, and re-evaluating strategy
5.6 Aligning actions with intentions
5.7 Prioritizing creativity and innovation in your work
5.8 Planning and executing your career path
5.9 Scaling solutions

Example: The Training our Future Ocean Leaders program at the University of British Columbia

With the support from Canada's Natural Sciences and Engineering Research Council, the "Ocean Leaders" program prepares graduate students and postdoctoral fellows to translate technical knowledge into management and policy innovation for the marine environment. The spine of the program is a two-semester "Grand Challenges" course, in which students work with community partners to take a local marine issue from the research and analysis stage through to the response and action stage. This learning-by-doing approach gives students real-world experience in the strategic thinking, planning, and collaboration skills necessary to solve the multifaceted challenges facing the world's oceans. In the research semester, student groups work with relevant community partners and the program faculty to identify research needs, and then complete a written scholarly research project (*skills 5.2, 5.4*). In the action semester, the groups develop and document a public or policy legacy product that addresses the previously-identified pressing challenge at an appropriate scale (*skills 5.6, 5.7, 5.9*). Project work is supported by workshops on collective leadership (*skill 5.1*), scenario planning (*skills 5.2, 5.4*), risk analysis and decision-making (*skill 5.3*), organizing for change (*skills 5.5, 5.9*), and entrepreneurial thinking (*skills 5.7, 5.8*). For example, in the entrepreneurial thinking workshop, students learn to arrive at well-supported policy recommendations through stakeholder mapping and value proposition development activities. Legacy products to date include guidance to a local First Nations community on marine spatial planning and a report for municipal government on improving coastal water quality. After the course ends, students can choose to continue working with community partners whose goals match their own with the support of undergraduate research assistants hired by the program (*skill 5.6*).

6. Working Productively and Efficiently

Sustainability leaders must be effective managers, networkers, and professionals with skills that are above and beyond their topical areas of expertise to ultimately get things done. The ability to make the most of what you have, while advocating for what more you need, is critical when balancing competing priorities and demands, contending with resource scarcity, and generating support for initiatives and ideas. LTS should foster abilities that facilitate not only what sustainability problems are addressed, but also how leaders-in-training will ultimately work to address them.

Associated Skills:

6.1 Curating your workload to have an impact
6.2 Organizing and strategizing personal priorities, boundaries, and progress
6.3 Managing time
6.4 Managing people
6.5 Managing projects
6.6 Managing finances
6.7 Understanding and navigating your organization
6.8 Leveraging assets, networks, relationships, and resources
6.9 Advocating for one self and others

Example: The Graduate Pursuit Program at the National Socio-Environmental Synthesis Center (SESYNC), University of Maryland, College Park

The Graduate Pursuit program is a cohort-based research opportunity open to PhD students from diverse fields and universities around the world. The program is designed to provide students with genuine interdisciplinary collaboration and scientific leadership experiences outside of their doctoral programs. To apply, students must first assemble a diverse team of peers—with the possibility of also recruiting external experts—who together have the skills and expertise needed to address a socio-environmental question or topic of interest (*skill 6.8*). Student teams must refine and articulate their collective idea as a compelling research proposal, in line with the mission and conditions of SESYNC (*skills 6.7, 6.9*). During the proposal development stage, students negotiate competing interests and views into a cohesive end-product while also planning roles and responsibilities around individuals' needs and availability (*skills 6.2, 6.9*). Meeting the demands of one's own research as well as the expectations of SESYNC and one's interdisciplinary team requires that students learn to balance multiple commitments as well as design research projects that equally maximize meaningful outcomes and feasibility (*skills 6.1, 6.2*). With access to a diverse suite of support services, teams are treated as independent scholars with full control over their research direction, but also with full responsibility for maintaining momentum, interacting effectively, and delivering outcomes within a project's 18- to24-month timeline (*skills 6.3, 6.4, 6.5*). Students must manage the financial support they receive in the forms of travel, lodging, and meal costs for three in-person meetings at SESYNC; open-access publication fees; and an honorarium upon program completion (*skill 6.6*).

7.    Making Your Work Matter

Sustainability leaders should know how to cultivate relationships and mutual understanding that will help push toward meaningful solutions to important problems. Despite solutions requiring input and buy-in from diverse stakeholders and decision-makers, translational skills are often not prioritized in graduate education. LTS that builds capacities among trainees to formulate ideas and solutions with relevant actors in mind and to interact with sectors that are necessary for facilitating desired change is ideal.

Associated Skills:

7.1    Designing your work for sustained impact
7.2    Relationship building and building meaningful networks
7.3    Engaging with the media
7.4    Engaging with and understanding needs of stakeholders
7.5    Engaging with and understanding needs of government and policymakers

Example: Environmental Impact Fellows Program at Duke University

The Environmental Impact Fellows Program is a professional development opportunity for doctoral students committed to making societal impacts throughout their careers. The program is designed to increase intentional leadership and is grounded in the following guiding principles initially developed by the Earth Leadership Program (formerly the Aldo Leopold Leadership Program): have empathy, be intentional, believe in lifelong learning, and be a systems thinker. The program has a two-pronged self-awareness training that emphasizes appreciating alternative value systems, cultivating empathy, and articulating personal value systems. One facet of the training is based on the Aspen Institute Executive Leadership program and is focused on readings and discussions intended to explore social issues that are key to having an impact. The other facet of the training is run by Barefoot Consulting, which shares a suite of tools to understand thinking styles, other points of view, getting buy-in, and mind mapping (*skills 7.2, 7.4*). Students participate in a communication training focused on written, oral, visual, and social media approaches for communicating science and answering controversial questions in order to cultivate the skills to engage with non-scientists (*skill 7.3*). They also participate in strategic planning for careers training, which teaches students how to build professional networks, set personal career goals,

and explore alternative career paths, including practice of informational interviews with stakeholders that work outside of academia (*skills 7.1, 7.2*). Finally, an engagement training focuses on learning how science informs public and private sector policy and structuring pathways to engage these audiences on science issues. This includes understanding how science gets integrated into public policy; identifying key audiences, stakeholders, communities, and decision makers; matching science advice to the context and needs of the target audience; and building relationships with decision-makers (*skills 7.1, 7.2, 7.4, 7.5*). Combined, the program provides opportunities for students to be introspective and explicit about their visions and goals for their research, the relationships and collaborations they build, and who will benefit from the outcomes of their work.

*4.2. Network Map*

We paired the release of the LTS aptitudes framework and curricular database with an online network map of ANGLES members and affiliates ("ANGLES Network Map" https://anglesnetwork.com/angles-map-page/). The goals for the network map were to make our own LTS program models—including how we each interpret and implement the LTS aptitudes and skills described above—more transparent and replicable. The network map also aimed to facilitate increased connections and the sharing of expertise between actors in the LTS space. The final network map displays programmatic information for a total of 35 categories (see Appendix C). The final categories are all those that meeting attendees agreed are the most relevant to understanding, borrowing from, or replicating existing LTS models from logistical and administrative standpoints. Those categories include, but are not limited to: staffing considerations, program scale and scope, funding sources, cost and or/funds available to students, time commitment, student eligibility, credentialing status, points of contact, and program goals. Other attributes include any evaluation methods used, recruitment methods, application and selection process, external partners, and mentorship models. The full range of network map categories, category response types, and category definitions are provided in Appendix C. At the time of writing, the continually growing network map hosts information for 18 LTS programs across the US and Canada.

## 5. Discussion

Ultimately, this multi-day collaborative endeavor was an exercise in complementing the existing literature, locating the common threads between our programs, and negotiating a dozen different individual and programmatic experiences and views. One notable instance of negotiation is our decision to use the term "aptitudes" as a neutral alternative for the more commonly employed language of "competencies" or "capacities." Though all three are used interchangeably in the literature, "competencies" and "capacities" meant different things to different people at the meeting. It was important to the group to select a word that suggested skills that can be acquired at varying ability levels. In other words, the group recognized that "competence" suggests that there is an endpoint; a point when you are "competent". In our model, we believe these are life-long skills that practitioners continue to grow.

One of the most important characteristics of the aptitudes and suite of nested skills is that they be considered, taught, and learned in conjunction with one another as opposed to in a piecemeal fashion. As a group, we agreed that when it comes to training the next generation of sustainability leaders, it is insufficient to pick and choose who is taught what. A good example from our conversations was the widespread agreement that a sustainability leader must not only be a good communicator, but a communicator who also possesses a broadly informed worldview, empathy, expert listening skills and who can serve in a "boundary spanning" role between diverse knowledges and communities [44]. Similarly, simply being able to ask good research questions is insufficient without a larger perspective around asking questions worth knowing and in light of real-world stakeholder needs.

As such, it was at times difficult to assign a hierarchical framework to skills and aptitudes. The group wanted to see skills interconnected across multiple aptitudes, for all participants felt strongly that each aptitude required some or all of the skills that were listed. This was particularly true for JEDI. The group engaged in a lengthy and robust discussion about whether or not it would be more appropriate to "lump" this topic as a standalone aptitude or to split it out across the others since it underpins myriad sustainability problems and is integral to lasting solutions. In the end, the group called attention to the importance of JEDI by making it its own aptitude, acknowledging that any attempt to articulate specific skills would inherently exclude other important skills. Nonetheless, by treating the aptitudes and skills as interlocking and inseparable, we highlighted the need to infuse JEDI-based actions and considerations across all dimensions of the framework.

Inadequate LTS, whether it be due to a lack of prioritization or uncertainties around implementation, is a systemic gap that plagues even the most highly-resourced institutions. Yet it is critical for future sustainability leaders to possess the knowledge, skills, and aptitudes central to LTS, such as collaborating and communicating, that will allow them to tackle increasingly complex sustainability problems [15]. We share the view that we must expand LTS to as many graduate students as possible, as quickly as possible, in order to better match the rapid pace at which our world is changing [1,10]. We also believe that the higher education system can play a key role in fostering the next generation of leadership for sustainability [45].

To get there, we must minimize burdens and barriers to entry for programs and faculty committed to this kind of training. In addition, we must solve the problem of limited interactions and learnings between LTS programs and those who run them. We know that competency-based training programs are capable of fast-tracking leaders in sustainability by boosting students' confidence, for example, and outfitting them with important, marketable skills [18,34]. However, as Pearson and colleagues note, "academics working in isolation, ignorant of the shape and scope of each other's contributions are most unlikely to deliver courses that equip their students with the knowledge, skills and values that sustainability requires" [46] (p. 184).

Through ANGLES, we seized an opportunity to extend synergies beyond our small network and to build on the important progress that has been achieved to date in several ways. Regarding our first goal of expanding and building consensus around the myriad competencies proposed to date, we built on previous work by delineating, defining, and describing the aptitudes that we believe effective sustainability leadership most demands. The skills and aptitudes we put forth are oriented around both leadership thinking and practice, given the mismatches found between what sustainability leadership programs tend to emphasize (e.g., content knowledge and systems thinking) and the interactive, interpersonal skills that students require most during their careers and leadership roles [24]. Our list of LTS aptitudes and skills complements and, at times, overlaps with sustainability competencies published elsewhere. For example, McGreavy et al. point out the importance of developing communication skills [3]; Rieckmann highlight how empathy is a critical sustainability leadership trait [32]; and Frisk and Larson note the need to train sustainability leaders in incorporating a diversity of perspectives [5]. Conflict resolution and concepts of justice are also among the skills that other sustainability-oriented frameworks share [22,47].

However, replicating existing frameworks was not a viable option for our bottom-up approach to integrating what we, in our varied programs, already know and do. Previous work on sustainability competencies have not prioritized leadership or have not been particularly useful for new and developing programs, given that calls for guidance are ongoing. Moreover, leadership theory and practice that is not specifically tailored to sustainability is not suited to navigating the realities of twenty-first century sustainability issues [48], because they were either designed around a particular field when today's graduate students require an interdisciplinary approach [7]; were focused on preparing individuals for academic careers when many graduate students will no longer take this

path; or were representative of hierarchical and perhaps even stereotypical modes of leadership that graduate students would likely find outdated and unrelatable.

In the end, our framework stands out as an amalgamation of a diverse community of practice specifically focused on training graduate students in leadership for sustainability. No single program represented at the meeting is a perfect example of this framework for LTS aptitude development. Our programs teach across many different aptitudes, yet are realistically limited by funds, time, and staffing capacities. As such, the aptitudes framework is not a proof of concept for any one existing model. Rather, we present it as a best-case scenario for LTS moving forward, which others can draw upon in program design. It is a result of us asking ourselves: what would be the sum of our many parts? How should future sustainability leaders be prepared, and how can we piece ourselves together to provide guidance around what that might ideally look like?

Our second goal, of providing clear guidance and grounded examples of how to operationalize what can be vague recommendations, allowed us to move beyond adding another set of competencies to a crowded domain and instead continue to bridge the gap between theory and practice via the conveyance of explicit skills [21]. Recommendations for what should be done in order to build proficiencies in sustainability leadership often fail to sufficiently translate into concrete, easily accessible, and tested resources and guidance [20]. Furthermore, demonstrating competencies in action remains uncommon despite sustainability requiring both knowledge and practice [4]. By synthesizing our programs' diverse approaches, insights, and expertise into an extensive, searchable online database, we model and share how a diverse suite of programs cultivates critical aptitudes for real-world graduate student trainees. By joining those who with similar goals who came before us (e.g., [6,9,18,34]), our hope is that such efforts will become the rule of the field, not the exception.

The simultaneous development of the aptitudes framework and database with an online, interactive network map brought to life our third goal of overcoming the general lack of communication and exchange between sustainability-oriented programs and the subsequent challenges that arise [23,46]. Facilitating access to people and programmatic details, including what needs to be in place for aptitude- and skill-building to occur, can help similar programs to launch and succeed, especially at under-resourced institutions or in settings where non-expert instructors are expected to implement such training [19]. Program leaders cannot be experts in everything that tomorrow's sustainability leaders need to know but should be able to easily identify and connect with a community of peers with diverse knowledge bases and a willingness to share. At the same time that it highlights and connects key programs and actors across the LTS landscape, this resource can additionally encourage the discovery of who and what is currently missing from LTS conversations and communities. In that sense, we view this as a living resource with much to be added and much untapped potential to transform our individual assets in order to achieve heightened collective impact.

We offer these products as the starting points for others to borrow from, re-shape, or be inspired by as they will, for there is no one formula for LTS. Rather, the resources presented here can help others pick and choose what pathways, contacts, and approaches align best with their own unique contexts and goals. Products can also showcase potential strategies for overcoming some of the most basic issues endangering the growth and longevity of LTS programs. Issues include, but are far from limited to, how to allocate staff, attract applicants, develop skills curricula without "reinventing the wheel", and meet the vast array of student needs without possessing each and every aptitude's expertise, either by borrowing from one another's work or identifying other experts to bring in.

Nonetheless, we are aware that LTS remains rare and expensive, and is often available to graduate students at only the most well-resourced institutions. We further recognize that the outcomes presented here are not solutions to many of the challenges facing sustainability leadership in higher education, such as lack of institutional interest and insufficient funding [49]. Outcomes are further limited in that their initial conceptualization

was based solely on the 13 individuals who were invited to and able to participate in the 2020 meeting. We have, however, since entered the next phase of the work wherein contributions from additional LTS professionals and institutions to the curricular database and network map will continue to test and expand the relevance of the framework and outcomes and modify them as needed.

## 6. Conclusions

Achieving a socially, economically, and environmentally sustainable future necessitates a different type of leadership; one we believe can and should be learned by as many as possible. Our recommendations and outcomes synthesize, for the first time, the programmatic experiences and expertise of nearly a dozen LTS programs and program leaders with previously identified gaps and calls for practical, concrete, and effective guidance. By translating programmatic philosophies, resources, and activities into an actionable framework and open-access tools available to LTS providers and learners alike, we provide a transparent, far-ranging, and more accessible look at Leadership Training for graduate students in Sustainability (LTS). We further offer a diverse array of tried and tested paths for others to learn from, add to, and adapt as their own.

Designing a program is hard work; designing one that matches and is capable of cultivating sustainability leaders in an era of rapidly change and growing complexity is even more difficult and fraught with obstacles. In taking these steps, we aimed to overcome some of the limitations, fragmentations, and uneven distributions that encumber the training of future sustainability leaders. We also aimed to accelerate the critical transition from identifying and describing sustainability competencies to modeling how they can be taught and learned in the context of leadership for sustainability. We encourage others to build on this work by continuing to blend conceptual insights for LTS with practical, implementable guidance. That, combined with greater attention to and support for LTS in higher education writ large, will better meet the needs and demands of this increasingly essential training.

**Author Contributions:** Conceptualization, N.M. and A.R.W.; methodology, A.R.W., N.M., K.C., S.D., R.J.H., C.J., M.K., L.O., K.R., L.S., D.H.W. and A.Y.; validation, A.R.W., N.M., K.C., S.D., R.J.H., C.J., M.K., L.O., K.R., L.S., D.H.W. and A.Y.; data curation, N.M. and A.R.W.; writing—original draft preparation, N.M. and A.R.W.; writing—review and editing, A.R.W., N.M., K.C., S.D., R.J.H., C.J., M.K., L.O., K.R., L.S., D.H.W. and A.Y.; funding acquisition, N.M. and A.R.W. All authors have read and agreed to the published version of the manuscript.

**Funding:** This work was supported by the US National Socio-Environmental Synthesis Center (SESYNC) under funding received from the National Science Foundation DBI-1639145.

**Institutional Review Board Statement:** Not applicable.

**Informed Consent Statement:** Not applicable.

**Data Availability Statement:** Data are available at https://anglesnetwork.com/angles-map-page/.

**Acknowledgments:** We thank Jonathan Kramer for facilitating the 2020 meeting and SESYNC for hosting the co-authors. We thank Ryan Deming and Micha Bennett for website development and figure creation. And we thank the reviewers for their useful recommendations and insights.

**Conflicts of Interest:** The authors declare no conflict of interest.

## Appendix A

List of aptitudes and associated skills

1.  Fostering Justice, Equity, Diversity, and Inclusion (JEDI) Associated Skills:
    1.1   Understanding the landscape of JEDI
    1.2   Appreciating the why of JEDI
    1.3   Approaching your work and making decisions informed by JEDI
    1.4   Fostering belonging & empowering others

2.  Building Emotional Intelligence Associated Skills:

    2.1 Building & maintaining personal integrity
    2.2 Recognizing your own strengths & weaknesses
    2.3 Recognizing your own values & motivations
    2.4 Having an empathetic mindset
    2.5 Valuing other's strengths, weaknesses, & values
    2.6 Aligning your personal values & intent with the actions & strategy you choose
    2.7 Rebounding from failure
    2.8 Identifying your leadership style

3.  Collaborating for Impact Associated Skills:

    3.1 Understanding & applying conflict resolution & negotiation skills
    3.2 Understanding team interactions & establishing shared visions, norms, processes, & trust
    3.3 Synthesizing and drawing connections among disparate ideas, information, theories, methods, evidence, & bodies of work
    3.4 Appreciating diverse views, priorities, values, & epistemologies
    3.5 Leveraging differences for improved lines of inquiry & problem solving
    3.6 Empowering others by recognizing their skills & expertise
    3.7 Co-developing & delivering outcomes

4.  Communicating and Engaging Associated Skills:

    4.1 Asking good questions and having good conversations
    4.2 Listening well and active listening
    4.3 Giving and receiving feedback
    4.4 Offering opposing points of view respectfully
    4.5 Knowing your audience and tailoring your communication accordingly
    4.6 Effectively presenting & conveying information

5.  Strategic Thinking and Planning Associated Skills:

    5.1 Comfort with and straddling the frontiers of ambiguity
    5.2 Brainstorming, visioning, & scenario planning
    5.3 Risk & decision analysis, and decision-making under uncertainty
    5.4 Seeing the big picture & thinking at the systems level
    5.5 Evaluating, adapting, & re-evaluating strategy
    5.6 Aligning actions with intentions
    5.7 Prioritizing creativity & innovation in your work
    5.8 Planning & executing your career path
    5.9 Scaling solutions

6.  Working Productively and Efficiently Associated Skills:

    6.1 Curating your workload to have an impact
    6.2 Organizing & strategizing personal priorities, boundaries, & progress
    6.3 Managing time
    6.4 Managing people
    6.5 Managing projects
    6.6 Managing finances
    6.7 Understanding & navigating your organization
    6.8 Leveraging assets, networks, relationships, & resources
    6.9 Advocating for one self & others

7.  Making Your Work Matter Associated Skills:

    7.1 Designing your work for sustained impact
    7.2 Relationship building and building meaningful networks
    7.3 Engaging with the media
    7.4 Engaging with & understanding needs of stakeholders
    7.5 Engaging with & understanding needs of government & policymakers

**Appendix B**

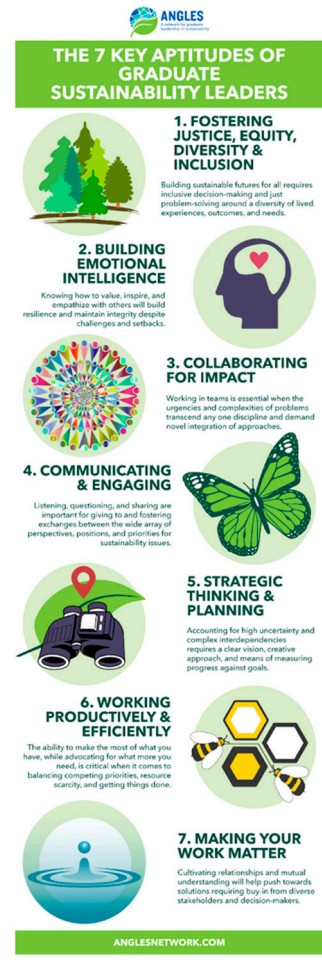

**Figure A1.** The 7 Key aptitudes of Graduate Sustainability Leaders. (Available online: https: //anglesnetwork.com/angles-aptitudes).

## Appendix C

**Table A1.** Data fields identified at the February 2020 meeting that were included in the final web version of the network map.

| Variable | Response Type | Definition |
|---|---|---|
| Institution | Short Answer | The academic institution or organization with which the graduate sustainability leadership training program is affiliated |
| Department/Institute/Unit | Short Answer | The institutional or organizational unit under which the program is housed |
| Program Title | Short Answer | The full title of the program |
| Program URL | Short Answer | Permanent link to program website |
| Country | Drop down (options: United States, Canada, Other) | The country in which program is based |
| State/Province | Drop down (options: states and provinces) | The state or province in which the program is based |
| City | Short answer | The city in which the program is based |
| Does the program focus on sustainability? | Drop down (options: yes/no) | |
| Does the program target graduate students? | Drop down (options: yes/no) | |
| Does the program feature leadership training? | Drop down (options: yes/no) | |
| Program Mission | Short answer | What the program strives to achieve, or why the program was founded |
| Year Initiated | Short answer | The calendar year program was founded |
| Currently Active | Drop down (options: yes/no) | Is the program still operating? |
| Program Scale | Drop down (options: department, campus, regional, national, international) | The operational reach of the program, or the level at which the program attracts applicants |
| Participant Model | Short answer | The main way participants interact with the program and/or each other |
| Participant Eligibility | Short answer | The criteria that determine whether an applicant is eligible to participate |
| Time Commitment | Short answer | The average amount of time participants should expect to dedicate |
| Participation Requirements | Short answer | The programmatic expectations for those involved |
| Application Frequency | Drop down (options: none, rolling, semesterly, yearly) | The frequency with which the program invites applications for the primary participant model |
| Application Process | Short answer | The process by which applicants apply to and are selected by the program for involvement |
| Participant Costs | Drop down (options: yes/no) | Any costs incurred to those involved? |
| Participant Funding/Support | Drop down (options: yes/no) | Any funds and/or non-monetary resources made available to participants? |
| Types of Participant Support | Short answer | The amount of funds made available to participants (if applicable) and/or the type(s) of non-monetary resources |
| Mentorship | Short answer | The availability and/or style of mentorship made available to those in the program |
| Participant Counts | Short answer | The number of participants involved in each application or activity cycle |
| Program Funder(s) | Short answer | The main sources of funding for the program and its activities |
| External Program Partners | Short answer | The extra-program organizations, experts, professionals, etc. with which the program regularly partners for activities, trainings, and/or offerings |
| Program Lead(s) | Short answer | The managers and/or directors of the program |
| Point(s) of Contact | Short answer | The person in charge of corresponding on behalf of the program |
| Contact Email(s) | Short answer | The point of contact's email address |
| Program Staff | Drop down (options: 1–10) | The number of full time employees that staff the program |
| Credit Bearing or Credentialing | Drop down (options: yes/no) | Do program participants earn credits or formal credentials from participating? |

**Table A1.** *Cont.*

| Variable | Response Type | Definition |
|---|---|---|
| Program Recruitment Strategies | Short answer | The main ways the program advertises its opportunities to potential applicants |
| Program Evaluation(s) Conducted | Drop down (options: yes/no) | Does the program conduct regular evaluations, or had an evaluation performed at some point? |
| Type(s) of Evaluation(s) Conducted | Short answer | The manner in which the evaluation is conducted |

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
