# Peer review of "Integrating Programmatic Expertise from across the US and Canada to Model and Guide Leadership Training for Graduate Students in Sustainability"

_sustainability, doi:10.3390/su13168950_

Round 1
Reviewer 1 Report
The paper is composed well. In addition to the framework of seven key aptitudes of graduate sustainability leaders, a model could be created, and a basic theoretical formulation.
Reviewer 2 Report
First of all, I would like to thank you for the opportunity to review the article entitled "Integrating Programmatic Expertise from Across the US and Canada to Model and Guide Leadership Training for Graduate Students in Sustainability.
The article is relevant to the scientific community. However, there are aspects that need to be addressed by the authors. These are detailed below:
- The citation and referencing of citations in the text and in the bibliography section does not conform to the journal's formatting and template guidelines. Authors should change this throughout the manuscript.
- In the introduction the authors talk about the competences needed for the education of students and the challenges of society. I understand that these skills are related to leadership, but the authors do not provide a definition of leadership. It would be important for the authors to indicate what it is, as a step before talking about the skills and abilities it requires or includes.
- At the end of the literature review, the authors indicate what this manuscript offers. However, they do not indicate the objective of the research. At least not clearly and concisely.
How was the contact with nearly 2,500 graduate students or other scholars through LTS initiatives? Did you work with all of them in the same way? The authors indicate more theory about their framework, but do not provide specifics about the method used in this section. How was the analysis of the data? Validity and reliability?
The presentation of the results needs to be improved. Right now they indicate 7 areas that were worked on or improved, providing what it means for their course. Also indicate skills associated with this area. I wonder if that is a gain for the course or an attempt by the theoretical authors to make it so. I encourage the authors to make a table in which they present the areas to be worked on and detail the results obtained for each of them.
Appropriate discussion.
Conclusions. The authors indicate that this work will help to reduce the limitations found in other training programmes. However, what are the limitations of this study and this programme?
Reviewer 3 Report
The article presents a topic that has a great potential for development, but at this stage it is insufficiently exposed in terms of clarity. Reading the text is difficult. The abstract can be improved in terms of structure, so as to provide a clearer picture of what will be presented later in the paper.
The introduction is well organized and quite clear, as well as the review of the literature in the field, but there is a confusion about the methodology. Instead of presenting the research methodology, descriptions are made of some networks, important for the topic, but which can also be framed as text in the previous part of the article.
On the other hand, the results section presents information that should appear in the methodology part.
The Results Section does not present the research output, but draws some methodological lines regarding the study, without exposing the concrete results obtained, respectively their processing. Or, in other words, the results are presented purely theoretically, without providing evidence of the study's application. The discussion part should include the results of one's own research, currently presenting approaches from the literature, which would have its place only in the literature review part.
Reviewer 4 Report
I appreciate the article submitted for review very positively. The article is very valuable in terms of cognition. I can conclude that it allows to deepen my current knowledge in relation to the issues of sustainable development. Congratulations on your selection of issues.
Round 2
Reviewer 2 Report
First of all, I would like to congratulate the authors for the changes made. I am aware that there were many changes, but they have certainly greatly improved the quality of the manuscript. The new version of the manuscript has exceeded my expectations.
Reviewer 3 Report
The authors managed to improve the paper and they addressed every observation. Congratulations and best regards!